# Using Self-Supervised Learning Can Improve Model Robustness and Uncertainty

**Dan Hendrycks**
UC Berkeley
hendrycks@berkeley.edu

**Mantas Mazeika**[*]
UIUC
mantas3@illinois.edu

**Saurav Kadavath**[*]
UC Berkeley
sauravkadavath@berkeley.edu

**Dawn Song**
UC Berkeley
dawnsong@berkeley.edu

## Abstract

Self-supervision provides effective representations for downstream tasks without requiring labels. However, existing approaches lag behind fully supervised training and are often not thought beneficial beyond obviating or reducing the need for annotations. We find that self-supervision can benefit robustness in a variety of ways, including robustness to adversarial examples, label corruption, and common input corruptions. Additionally, self-supervision greatly benefits out-of-distribution detection on difficult, near-distribution outliers, so much so that it exceeds the performance of fully supervised methods. These results demonstrate the promise of self-supervision for improving robustness and uncertainty estimation and establish these tasks as new axes of evaluation for future self-supervised learning research.

## 1  Introduction

Self-supervised learning holds great promise for improving representations when labeled data are scarce. In semi-supervised learning, recent self-supervision methods are state-of-the-art [Gidaris et al., 2018, Dosovitskiy et al., 2016, Zhai et al., 2019], and self-supervision is essential in video tasks where annotation is costly [Vondrick et al., 2016, 2018]. To date, however, self-supervised approaches lag behind fully supervised training on standard accuracy metrics and research has existed in a mode of catching up to supervised performance. Additionally, when used in conjunction with fully supervised learning on a fully labeled dataset, self-supervision has little impact on accuracy. This raises the question of whether large labeled datasets render self-supervision needless.

We show that while self-supervision does not substantially improve accuracy when used in tandem with standard training on fully labeled datasets, it can improve several aspects of model robustness, including robustness to adversarial examples [Madry et al., 2018], label corruptions [Patrini et al., 2017, Zhang and Sabuncu, 2018], and common input corruptions such as fog, snow, and blur [Hendrycks and Dietterich, 2019]. Importantly, these gains are masked if one looks at clean accuracy alone, for which performance stays constant. Moreover, we find that self-supervision greatly improves out-of-distribution detection for difficult, near-distribution examples, a long-standing and underexplored problem. In fact, using self-supervised learning techniques on CIFAR-10 and ImageNet for out-of-distribution detection, we are even able to *surpass fully supervised methods*.

These results demonstrate that self-supervision need not be viewed as a collection of techniques allowing models to catch up to full supervision. Rather, using the two in conjunction provides strong regularization that improves robustness and uncertainty estimation even if clean accuracy does not

---

[*]Equal Contribution.

change. Importantly, these methods can improve robustness and uncertainty estimation without requiring larger models or additional data [Schmidt et al., 2018, Kurakin et al., 2017]. They can be used with task-specific methods for additive effect with no additional assumptions. With self-supervised learning, we make tangible progress on adversarial robustness, label corruption, common input corruptions, and out-of-distribution detection, suggesting that future self-supervised learning methods could also be judged by their utility for uncertainty estimates and model robustness. Code and our expanded ImageNet validation dataset are available at https://github.com/hendrycks/ss-ood.

## 2   Related Work

**Self-supervised learning.**   A number of self-supervised methods have been proposed, each exploring a different pretext task. Doersch et al. [2015] predict the relative position of image patches and use the resulting representation to improve object detection. Dosovitskiy et al. [2016] create surrogate classes to train on by transforming seed image patches. Similarly, Gidaris et al. [2018] predict image rotations (Figure 1). Other approaches include using colorization as a proxy task [Larsson et al., 2016], deep clustering methods [Ji et al., 2018], and methods that maximize mutual information [Hjelm et al., 2019] with high-level representations [van den Oord et al., 2018, Hénaff et al., 2019]. These works focus on the utility of self-supervision for learning without labeled data and do not consider its effect on robustness and uncertainty.

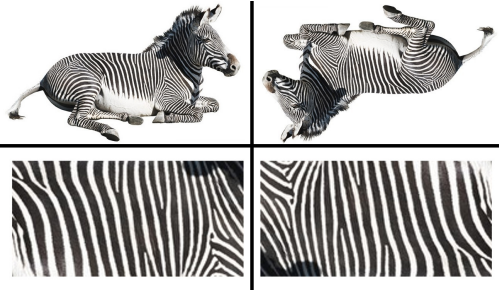

Figure 1: Predicting rotation requires modeling shape. Texture alone is not sufficient for determining whether the zebra is flipped, although it may be sufficient for classification under ideal conditions. Thus, training with self-supervised auxiliary rotations may improve robustness.

**Robustness.**   Improving model robustness refers to the goal of ensuring machine learning models are resistant across a variety of imperfect training and testing conditions. Hendrycks and Dietterich [2019] look at how models can handle common real-world image corruptions (such as fog, blur, and JPEG compression) and propose a comprehensive set of distortions to evaluate real-world robustness. Another robustness problem is learning in the presence of corrupted labels [Nettleton et al., 2010, Patrini et al., 2017]. To this end, Hendrycks et al. [2018] introduce Gold Loss Correction (GLC), a method that uses a small set of trusted labels to improve accuracy in this setting. With high degrees of label corruption, models start to overfit the misinformation in the corrupted labels [Zhang and Sabuncu, 2018, Hendrycks et al., 2019a], suggesting a need for ways to supplement training with reliable signals from unsupervised objectives. Madry et al. [2018] explore adversarial robustness and propose PGD adversarial training, where models are trained with a minimax robust optimization objective. Zhang et al. [2019] improve upon this work with a modified loss function and develop a better understanding of the trade-off between adversarial accuracy and natural accuracy.

**Out-of-distribution detection.**   Out-of-distribution detection has a long history. Traditional methods such as one-class SVMs [Schölkopf et al., 1999] have been revisited with deep representations [Ruff et al., 2018], yielding improvements on complex data. A central line of recent exploration has been with out-of-distribution detectors using supervised representations. Hendrycks and Gimpel [2017] propose using the maximum softmax probability of a classifier for out-of-distribution detection. Lee et al. [2018] expand on this by generating synthetic outliers and training the representations to flag these examples as outliers. However, Hendrycks et al. [2019b] find that training against a large and diverse dataset of outliers enables far better out-of-distribution detection on unseen distributions. In these works, detection is most difficult for near-distribution outliers, which suggests a need for new methods that force the model to learn more about the structure of in-distribution examples.

## 3   Robustness

### 3.1   Robustness to Adversarial Perturbations

Improving robustness to adversarial inputs has proven difficult, with adversarial training providing the only longstanding gains [Carlini and Wagner, 2017, Athalye et al., 2018]. In this section, we demonstrate that auxiliary self-supervision in the form of predicting rotations [Gidaris et al., 2018] can

|                          | Clean | 20-step PGD | 100-step PGD |
|--------------------------|-------|-------------|--------------|
| Normal Training          | 94.8  | 0.0         | 0.0          |
| Adversarial Training     | 84.2  | 44.8        | 44.8         |
| + Auxiliary Rotations (Ours) | 83.5 | 50.4    | 50.4         |

Table 1: Results for our defense. All results use $\varepsilon = 8.0/255$. For 20-step adversaries $\alpha = 2.0/255$, and for 100-step adversaries $\alpha = 0.3/255$. More steps do not change results, so the attacks converge. Self-supervision through rotations provides large gains over standard adversarial training.

improve upon standard Projected Gradient Descent (PGD) adversarial training [Madry et al., 2018]. We also observe that self-supervision can provide gains when combined with stronger defenses such as TRADES [Zhang et al., 2019] and is not broken by gradient-free attacks such as SPSA [Uesato et al., 2018].

**Setup.** The problem of defending against bounded adversarial perturbations can be formally expressed as finding model parameters $\theta$ for the classifier $p$ that minimize the objective

$$\min_\theta \mathbb{E}_{(x,y) \sim \mathcal{D}} \left[ \max_{x' \in S} \mathcal{L}_{\text{CE}}(y, p(y \mid x'); \theta) \right] \quad \text{where} \quad S = \{x' : \|x - x'\| < \varepsilon\} \qquad (1)$$

In this paper, we focus on $\ell_\infty$ norm bounded adversaries. Madry et al. [2018] propose that PGD is "a universal first-order adversary." Hence, we first focus on defending against PGD. Let $\text{PGD}(x)$ be the $K^{\text{th}}$ step of PGD,

$$x^{k+1} = \Pi_S \left( x^k + \alpha \, \text{sign}(\nabla_x \mathcal{L}_{\text{CE}}(y, p(y \mid x^k); \theta))) \right) \quad \text{and} \quad x^0 = x + U(-\varepsilon, \varepsilon) \qquad (2)$$

where $K$ is a preset parameter which characterizes the number of steps that are taken, $\Pi_S$ is the projection operator for the $l_\infty$ ball $S$, and $\mathcal{L}_{\text{CE}}(y, p(y \mid x'); \theta)$ is the loss we want to optimize. Normally, this loss is the cross entropy between the model's softmax classification output for $x$ and the ground truth label $y$. For evaluating robust accuracy, we use 20-step and 100-step adversaries. For the 20-step adversary, we set the step-size $\alpha = 2/256$. For the 100-step adversary, we set $\alpha = 0.3/256$ as in [Madry et al., 2018]. During training, we use 10-step adversaries with $\alpha = 2/256$.

In all experiments, we use 40-2 Wide Residual Networks Zagoruyko and Komodakis [2016]. For training, we use SGD with Nesterov momentum of 0.9 and a batch size of 128. We use an initial learning rate of 0.1 and a cosine learning rate schedule Loshchilov and Hutter [2016] and weight decay of $5 \times 10^{-4}$. For data augmentation, we use random cropping and mirroring. Hyperparameters were chosen as standard values and are used in subsequent sections unless otherwise specified.

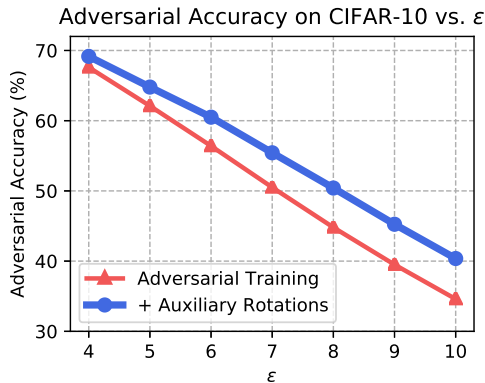

Figure 2: The effect of attack strength on a $\varepsilon = 8/255$ adversarially trained model. The attack strengths are $\varepsilon \in \{4/255, 5/255, \ldots, 10/255\}$. Since the accuracy gap widens as $\varepsilon$ increases, self-supervision's benefits are masked when observing the clean accuracy alone.

**Method.** We explore improving representation robustness beyond standard PGD training with auxiliary rotation-based self-supervision in the style of Gidaris et al. [2018]. In our approach, we train a classification network along with a separate auxiliary head, which takes the penultimate vector from the network as input and outputs a 4-way softmax distribution. This head is trained along with the rest of the network to predict the amount of rotation applied to a given input image (from 0°, 90°, 180°, and 270°). Our overall loss during training can be broken down into a supervised loss and a self-supervised loss

$$\mathcal{L}(x, y; \theta) = \mathcal{L}_{\text{CE}}(y, p(y \mid \text{PGD}(x)); \theta) + \lambda \mathcal{L}_{\text{SS}}(\text{PGD}(x); \theta). \qquad (3)$$

Note that the self-supervised component of the loss does not require the ground truth training label $y$ as input. The supervised loss does not make use of our auxiliary head, while the self-supervised loss

only makes use of this head. When $\lambda = 0$, our total loss falls back to the loss used for PGD training. For our experiments, we use $\lambda = 0.5$ and the following rotation-based self-supervised loss

$$\mathcal{L}_{\text{SS}}(x;\theta) = \frac{1}{4}\left[\sum_{r\in\{0^\circ,90^\circ,180^\circ,270^\circ\}}\mathcal{L}_{\text{CE}}(\texttt{one\_hot}(r), p_{\texttt{rot\_head}}(r \mid R_r(x));\theta)\right], \qquad (4)$$

where $R_r(x)$ is a rotation transformation and $\mathcal{L}_{\text{CE}}(x,r;\theta)$ is the cross-entropy between the auxiliary head's output and the ground-truth label $r \in \{0^\circ, 90^\circ, 180^\circ, 270^\circ\}$. In order to adapt the PGD adversary to the new training setup, we modify the loss used in the PGD update equation (2) to maximize both the rotation loss and the classification loss. In the Appendix, we find that this modification is optional and that the main source of improvement comes from the rotation loss itself. We report results with the modification here, for completeness. The overall loss that PGD will try to maximize for each training image is $\mathcal{L}_{\text{CE}}(y, p(y \mid x);\theta) + \mathcal{L}_{\text{SS}}(x;\theta)$. At test-time, the PGD loss does not include the $\mathcal{L}_{\text{SS}}$ term, as we want to attack the image classifier and not the rotation classifier.

**Results and analysis.** We are able to attain large improvements over standard PGD training by adding self-supervised rotation prediction. Table 1 contains results of our model against PGD adversaries with $K = 20$ and $K = 100$. In both cases, we are able to achieve a 5.6% absolute improvement over classical PGD training. In Figure 2, we observe that our method of adding auxiliary rotations actually provides larger gains over standard PGD training as the maximum perturbation distance $\varepsilon$ increases. The figure also shows that our method can withstand up to 11% larger perturbations than PGD training without any drop in performance.

In order to demonstrate that our method does not rely on gradient obfuscation, we attempted to attack our models using SPSA [Uesato et al., 2018] and failed to notice any performance degradation compared to standard PGD training. In addition, since our self-supervised method has the nice property of being easily adaptable to supplement other different supervised defenses, we also studied the effect of adding self-supervised rotations to stronger defenses such as TRADES [Zhang et al., 2019]. We found that self-supervision is able to help in this setting as well. Our best-performing TRADES + rotations model gives a 1.22% boost over standard TRADES and a 7.79% boost over standard PGD training in robust accuracy. For implementation details, see code.

## 3.2 Robustness to Common Corruptions

**Setup.** In real-world applications of computer vision systems, inputs can be corrupted in various ways that may not have been encountered during training. Improving robustness to these common corruptions is especially important in safety-critical applications. Hendrycks and Dietterich [2019] create a set of fifteen test corruptions and four validation corruptions common corruptions to measure input corruption robustness. These corruptions fall into noise, blur, weather, and digital categories. Examples include shot noise, zoom blur, snow, and JPEG compression.

We use the CIFAR-10-C validation dataset from Hendrycks and Dietterich [2019] and compare the robustness of normally trained classifiers to classifiers trained with an auxiliary rotation prediction loss. As in previous sections, we predict all four rotations in parallel in each batch. We use 40-2 Wide Residual Networks and the same optimization hyperparameters as before. We do not tune on the validation corruptions, so we report average performance over all corruptions. Results are in Figure 3.

**Results and analysis.** The baseline of normal training achieves a clean accuracy of 94.7% and an average accuracy over all corruptions of 72.3%. Training with auxiliary rotations maintains clean accuracy at 95.5% but increases the average accuracy on corrupted images by 4.6% to 76.9%. Thus, the benefits of self-supervision to robustness are masked by similar accuracy on clean images. Performance gains are spread across corruptions, with a small loss of performance in only one corruption type, JPEG compression. For glass blur, clean accuracy improves by 11.4%, and for Gaussian noise it improves by 11.6%. Performance is also improved by 8.9% on contrast and shot noise and 4.2% on frost, indicating substantial gains in robustness on a wide variety of corruptions. These results demonstrate that self-supervision can regularize networks to be more robust even if clean accuracy is not affected.

## 3.3 Robustness to Label Corruptions

**Setup.** Training classifiers on corrupted labels can severely degrade performance. Thus, several prior works have explored training deep neural networks to be robust to label noise in the multi-class

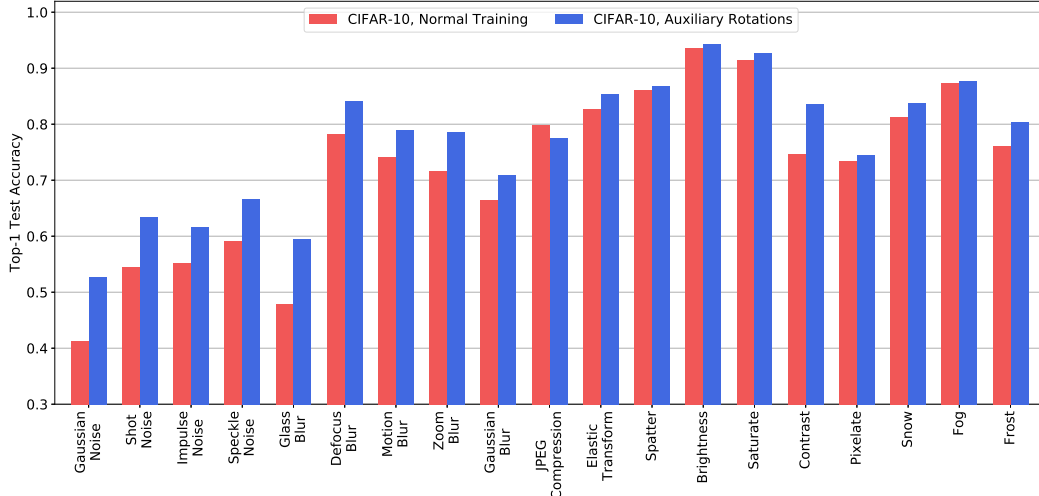

Figure 3: A comparison of the accuracy of usual training compared to training with auxiliary rotation self-supervision on the nineteen CIFAR-10-C corruptions. Each bar represents an average over all five corruption strengths for a given corruption type.

classification setting Sukhbaatar et al. [2014], Patrini et al. [2017], Hendrycks et al. [2018]. We use the problem setting from these works. Let $x$, $y$, and $\widetilde{y}$ be an input, clean label, and potentially corrupted label respectively. Given a dataset $\widehat{\mathcal{D}}$ of $(x, \widetilde{y})$ pairs for training, the task is to obtain high classification accuracy on a test dataset $\mathcal{D}_{\text{test}}$ of cleanly-labeled $(x, y)$ pairs.

Given a cleanly-labeled training dataset $\widetilde{\mathcal{D}}$, we generate $\widetilde{\mathcal{D}}$ with a corruption matrix $C$, where $C_{ij} = p(\widetilde{y} = j \mid y = i)$ is the probability of a ground truth label $i$ being corrupted to $j$. Where $K$ is the range of the label, we construct $C$ according to $C = (1 - s)I_K + s11^{\mathsf{T}}/K$. In this equation, $s$ is the corruption strength, which lies in $[0, 1]$. At a corruption strength of 0, the labels are unchanged, while at a corruption strength of 1 the labels have an equal chance of being corrupted to any class. To measure performance, we average performance on $\mathcal{D}_{\text{test}}$ over corruption strengths from 0 to 1 in increments of 0.1 for a total of 11 experiments.

**Methods.** Training without loss correction methods or self-supervision serves as our first baseline, which we call *No Correction* in Table 2. Next, we compare to the state-of-the-art *Gold Loss Correction (GLC)* Hendrycks et al. [2018]. This is a two-stage loss correction method based on Sukhbaatar et al. [2014] and Patrini et al. [2017]. The first stage of training estimates the matrix $C$ of conditional corruption probabilities, which partially describes the corruption process. The second stage uses the estimate of $C$ to train a corrected classifier that performs well on the clean label distribution. The *GLC* assumes access to a small dataset of trusted data with cleanly-labeled examples. Thus, we specify the percent of amount of trusted data available in experiments as a fraction of the training set. This setup is also known as a semi-verified setting Charikar et al. [2017].

To investigate the effect of self-supervision, we use the combined loss $\mathcal{L}_{\text{CE}}(y, p(y \mid x); \theta) + \lambda\mathcal{L}_{\text{SS}}(x; \theta)$, where the first term is standard cross-entropy loss and the second term is the auxiliary rotation loss defined in Section 3.1. We call this *Rotations* in Table 2. In all experiments, we set $\lambda = 0.5$. Gidaris et al. [2018] demonstrate that predicting rotations can yield effective representations for subsequent fine-tuning on target classification tasks. We build on this approach and pre-train with the auxiliary rotation loss alone for 100 epochs, after which we fine-tune for 40 epochs with the combined loss.

We use 40-2 Wide Residual Networks [Zagoruyko and Komodakis, 2016]. Hyperparameters remain unchanged from Section 3.1. To select the number of fine-tuning epochs, we use a validation split of the CIFAR-10 training dataset with clean labels and select a value to bring accuracy close to that of *Normal Training*. Results are in Table 2 and performance curves are in Figure 4.

**Analysis.** We observe large gains in robustness from auxiliary rotation prediction. Without loss corrections, we reduce the average error by 5.6% on CIFAR-10 and 5.2% on CIFAR-100. This corresponds to an 11% relative improvement over the baseline of normal training on CIFAR-100

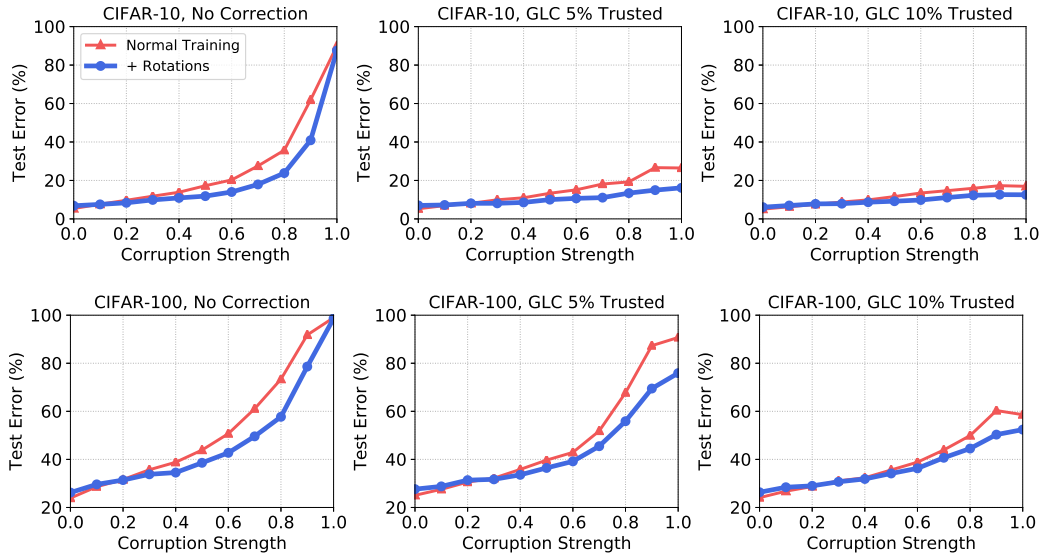

Figure 4: Error curves for label corruption comparing normal training to training with auxiliary rotation self-supervision. Auxiliary rotations improve performance when training without loss corrections and are complementary with the GLC loss correction method.

|                    | CIFAR-10 | | CIFAR-100 | |
|                    | Normal Training | Rotations | Normal Training | Rotations |
|--------------------|-----------------|-----------|-----------------|-----------|
| No Correction      | 27.4            | 21.8      | 52.6            | 47.4      |
| GLC (5% Trusted)   | 14.6            | 10.5      | 48.3            | 43.2      |
| GLC (10% Trusted)  | 11.6            | 9.6       | 39.1            | 36.8      |

Table 2: Label corruption results comparing normal training to training with auxiliary rotation self-supervision. Each value is the average error over 11 corruption strengths. All values are percentages. The reliable training signal from self-supervision improves resistance to label noise.

and a 26% relative improvement on CIFAR-10. In fact, auxiliary rotation prediction with no loss correction outperforms the GLC with 5% trusted data on CIFAR-100. This is surprising given that the GLC was developed specifically to combat label noise.

We also observe additive effects with the GLC. On CIFAR-10, the GLC with 5% trusted data obtains 14.6% average error, which is reduced to 10.5% with the addition of auxiliary rotation prediction. Note that doubling the amount of trusted data to 10% yields 11.6% average error. Thus, using self-supervision can enable obtaining better performance than doubling the amount of trusted data in a semi-supervised setting. On CIFAR-100, we observe similar complementary gains from auxiliary rotation prediction. Qualitatively, we can see in Figure 4 that performance degradation as the corruption strength increases is softer with auxiliary rotation prediction.

On CIFAR-100, error at 0% corruption strength is 2.3% higher with auxiliary rotation predictions. This is because we selected the number of fine-tuning epochs on CIFAR-10 at 0% corruption strength, for which the degradation is only 1.3%. Fine-tuning for longer can eliminate this gap, but also leads to overfitting label noise [Zhang and Sabuncu, 2018]. Controlling this trade-off of robustness to performance on clean data is application-specific. However, past a corruption strength of 20%, auxiliary rotation predictions improve performance for all tested corruption strengths and methods.

## 4 Out-of-Distribution Detection

Self-supervised learning with rotation prediction enables the detection of harder out-of-distribution examples. In the following two sections, we show that self-supervised learning improves out-of-distribution detection when the in-distribution consists in multiple classes or just a single class.

## 4.1 Multi-Class Out-of-Distribution Detection.

**Setup.** In the following experiment, we train a CIFAR-10 classifier and use it as an out-of-distribution detector. When given an example $x$, we write the classifier's posterior distribution over the ten classes with $p(y \mid x)$. Hendrycks and Gimpel [2017] show that $p(y \mid x)$ can enable the detection of out-of-distribution examples. They show that the maximum softmax probability $\max_c p(y = c \mid x)$ tends to be higher for in-distribution examples than for out-of-distribution examples across a range of tasks, enabling the detection of OOD examples.

We evaluate each OOD detector using the area under the receiver operating characteristic curve (AUROC) [Davis and Goadrich, 2006]. Given an input image, an OOD detector produces an anomaly score. The AUROC is equal to the probability an out-of-distribution example has a higher anomaly score than an in-distribution example. Thus an OOD detector with a 50% AUROC is at random-chance levels, and one with a 100% AUROC is without a performance flaw.

**Method.** We train a classifier with an auxiliary self-supervised rotation loss. The loss during training is $\mathcal{L}_{\text{CE}}(y, p(y \mid x)) + \sum_{r \in \{0°, 90°, 180°, 270°\}} \mathcal{L}_{\text{CE}}(\texttt{one\_hot}(r), p_{\texttt{rot\_head}}(r \mid R_r(x)))$, and we only train on in-distribution CIFAR-10 training examples. After training is complete, we score in-distribution CIFAR-10 test set examples and OOD examples with the formula $\text{KL}[U\|p(y \mid x)] + \frac{1}{4}\sum_{r \in \{0°, 90°, 180°, 270°\}} \mathcal{L}_{\text{CE}}(\texttt{one\_hot}(r), p_{\texttt{rot\_head}}(r \mid R_r(x)))$. We use the KL divergence of the softmax prediction to the uniform distribution $U$ since it combines well with the rotation score, and because Hendrycks et al. [2019b] show that $\text{KL}[U\|p(y \mid x)]$ performs similarly to the maximum softmax probability baseline $\max_c p(y = c \mid x)$.

The training loss is standard cross-entropy loss with auxiliary rotation prediction. The detection score is the KL divergence detector from prior work with a rotation score added to it. The rotation score consists of the cross entropy of the rotation softmax distribution to the categorical distribution over rotations with probability 1 at the current rotation and 0 everywhere else. This is equivalent to the negative log probability assigned to the true rotation. Summing the cross entropies over the rotations gives the total rotation score.

| Method | AUROC |
|---|---|
| Baseline | 91.4% |
| Rotations (Ours) | 96.2% |

Figure 5: OOD detection performance of the maximum softmax probability baseline and our method using self-supervision. Full results are in the Appendix.

**Results and Analysis.** We evaluate this proposed method against the maximum softmax probability baseline [Hendrycks and Gimpel, 2017] on a wide variety of anomalies with CIFAR-10 as the in-distribution data. For the anomalies, we select Gaussian, Rademacher, Blobs, Textures, SVHN, Places365, LSUN, and CIFAR-100 images. We observe performance gains across the board and report average AUROC values in Figure 5. On average, the rotation method increases the AUROC by 4.8%.

This method does not require additional data as in Outlier Exposure [Hendrycks et al., 2019b], although combining the two could yield further benefits. As is, the performance gains are of comparable magnitude to more complex methods proposed in the literature [Xie et al., 2018]. This demonstrates that self-supervised auxiliary rotation prediction can augment OOD detectors based on fully supervised multi-class representations. More detailed descriptions of the OOD datasets and the full results on each anomaly type with additional metrics are in the Appendix.

## 4.2 One-Class Learning

**Setup.** In the following experiments, we take a dataset consisting in $k$ classes and train a model on one class. This model is used as an out-of-distribution detector. For the source of OOD examples, we use the examples from the remaining unseen $k-1$ classes. Consequently, for the datasets we consider, the OOD examples are near the in-distribution and make for a difficult OOD detection challenge.

### 4.2.1 CIFAR-10

**Baselines.** One-class SVMs [Schölkopf et al., 1999] are an unsupervised out-of-distribution detection technique which models the training distribution by finding a small region containing most of the training set examples, and points outside this region are deemed OOD. In our experiment, OC-SVMs operate on the raw CIFAR-10 pixels. Deep SVDD [Ruff et al., 2018] uses convolutional networks to extract features from the raw pixels all while modelling one class, like OC-SVMs.

|  | OC-SVM | DeepSVDD | Geometric | RotNet | DIM | IIC | Supervised (OE) | Ours | Ours + OE |
|---|---|---|---|---|---|---|---|---|---|
| Airplane | 65.6 | 61.7 | 76.2 | 71.9 | 72.6 | 68.4 | 87.6 | 77.5 | 90.4 |
| Automobile | 40.9 | 65.9 | 84.8 | 94.5 | 52.3 | 89.4 | 93.9 | 96.9 | 99.3 |
| Bird | 65.3 | 50.8 | 77.1 | 78.4 | 60.5 | 49.8 | 78.6 | 87.3 | 93.7 |
| Cat | 50.1 | 59.1 | 73.2 | 70.0 | 53.9 | 65.3 | 79.9 | 80.9 | 88.1 |
| Deer | 75.2 | 60.9 | 82.8 | 77.2 | 66.7 | 60.5 | 81.7 | 92.7 | 97.4 |
| Dog | 51.2 | 65.7 | 84.8 | 86.6 | 51.0 | 59.1 | 85.6 | 90.2 | 94.3 |
| Frog | 71.8 | 67.7 | 82.0 | 81.6 | 62.7 | 49.3 | 93.3 | 90.9 | 97.1 |
| Horse | 51.2 | 67.3 | 88.7 | 93.7 | 59.2 | 74.8 | 87.9 | 96.5 | 98.8 |
| Ship | 67.9 | 75.9 | 89.5 | 90.7 | 52.8 | 81.8 | 92.6 | 95.2 | 98.7 |
| Truck | 48.5 | 73.1 | 83.4 | 88.8 | 47.6 | 75.7 | 92.1 | 93.3 | 98.5 |
| Mean | 58.8 | 64.8 | 82.3 | 83.3 | 57.9 | 67.4 | 87.3 | 90.1 | 95.6 |

Table 3: AUROC values of different OOD detectors trained on one of ten CIFAR-10 classes. Test time out-of-distribution examples are from the remaining nine CIFAR-10 classes. In-distribution examples are examples belonging to the row's class. Our self-supervised technique surpasses a fully supervised model. All values are percentages.

RotNet [Gidaris et al., 2018] is a successful self-supervised technique which learns its representations by predicting whether an input is rotated 0°, 90°, 180°, or 270°. After training RotNet, we use the softmax probabilities to determine whether an example is in- or out-of-distribution. To do this, we feed the network the original example (0°) and record RotNet's softmax probability assigned to the 0° class. We then rotate the example 90° and record the probability assigned to the 90° class. We do the same for 180° and 270°, and add up these probabilities. The sum of the probabilities of in-distribution examples will tend to be higher than the sum for OOD examples, so the negative of this sum is the anomaly score. Next, Golan and El-Yaniv [2018] (Geometric) predicts transformations such as rotations and whether an input is horizontally flipped; we are the first to connect this method to self-supervised learning and we improve their method. Deep InfoMax [Hjelm et al., 2019] networks learn representations which have high mutual information with the input; for detection we use the scores of the discriminator network. A recent self-supervised technique is Invariant Information Clustering (IIC) [Ji et al., 2018] which teaches networks to cluster images without labels but instead by learning representations which are invariant to geometric perturbations such as rotations, scaling, and skewing. For our supervised baseline, we use a deep network which performs logistic regression, and for the negative class we use Outlier Exposure. In Outlier Exposure, the network is exposed to examples from a real, diverse dataset of consisting in out-of-distribution examples. Done correctly, this process teaches the network to generalize to unseen anomalies. For the outlier dataset, we use 80 Million Tiny Images [Torralba et al., 2008] with CIFAR-10 and CIFAR-100 examples removed. Crucial to the success of the supervised baseline is our loss function choice. To ensure the supervised baseline learns from hard examples, we use the Focal Loss [Lin et al., 2017].

**Method.** For our self-supervised one-class OOD detector, we use a deep network to predict geometric transformations and thereby surpass previous work and the fully supervised network. Examples are rotated 0°, 90°, 180°, or 270° then translated 0 or ±8 pixels vertically and horizontally. These transformations are composed together, and the network has three softmax heads: one for predicting rotation ($\mathcal{R}$), one for predicting vertical translations ($\mathcal{T}_v$), and one for predicting horizontal translations ($\mathcal{T}_h$). Concretely, the anomaly score for an example $x$ is

$$\sum_{r \in \mathcal{R}} \sum_{s \in \mathcal{T}_v} \sum_{t \in \mathcal{T}_h} p_{\texttt{rot\_head}}(r \mid G(x)) + p_{\texttt{vert\_transl\_head}}(s \mid G(x)) + p_{\texttt{horiz\_transl\_head}}(t \mid G(x)),$$

where $G$ is the composition of rotations, vertical translations, and horizontal translations specified by $r$, $p$, and $q$ respectively. The set $\mathcal{R}$ is the set of rotations, and $p_{\texttt{rot\_head}}(r \mid \cdot)$ is the softmax probability assigned to rotation $r$ by the rotation predictor. Likewise with translations for $\mathcal{T}_v$, $\mathcal{T}_h$, $s$, $t$, $p_{\texttt{vert\_transl\_head}}$, and $p_{\texttt{horiz\_transl\_head}}$. The backbone architecture is a 16-4 WideResNet [Zagoruyko and Komodakis, 2016] trained with a dropout rate of 0.3 [Srivastava et al., 2014]. We choose a 16-4 network because there are fewer training samples. Networks are trained with a cosine learning rate schedule [Loshchilov and Hutter, 2016], an initial learning rate of 0.1, Nesterov momentum, and a batch size of 128. Data is augmented with standard cropping and mirroring. Our RotNet and supervised baseline use the same backbone architecture and training hyperparameters. When training our method with Outlier Exposure, we encourage the network to have uniform softmax responses on

out-of-distribution data. For Outlier Exposure to work successfully, we applied the aforementioned geometric transformations to the outlier images so that the in-distribution data and the outliers are as similar as possible.

Results are in Table 3. Notice many self-supervised techniques perform better than methods specifically designed for one-class learning. Also notice that our self-supervised technique outperforms Outlier Exposure, the state-of-the-art fully supervised method, which also requires access to out-of-distribution samples to train. In consequence, a model trained with self-supervision can surpass a fully supervised model. Combining our self-supervised technique with supervision through Outlier Exposure nearly solves this CIFAR-10 task.

### 4.2.2 ImageNet

**Dataset.** We consequently turn to a harder dataset to test self-supervised techniques. For this experiment, we select 30 classes from ImageNet Deng et al. [2009]. See the Appendix for the classes.

**Method.** Like before, we demonstrate that a self-supervised model can surpass a model that is fully supervised. The fully supervised model is trained with Outlier Exposure using ImageNet-22K outliers (with ImageNet-1K images removed). The architectural backbone for these experiments is a ResNet-18. Images are resized such that the smallest side has 256 pixels, while the aspect ratio is maintained. Images are randomly cropped to the size $224 \times 224 \times 3$. Since images are larger than CIFAR-10, new additions to the self-supervised method are possible. Consequently, we can teach the network to predict whether than image has been resized. In addition, since we should like the network to more easily learn shape and compare regions across the whole image, we discovered there is utility in self-attention [Woo et al., 2018] for this task. Other architectural changes, such as using a Wide *RevNet* [Behrmann et al., 2018] instead of a Wide ResNet, can increase the AUROC from 65.3% to 77.5%. AUROCs are shown in Table 4. Self-supervised methods outperform the fully supervised baseline by a large margin, yet there is still wide room for improvement on large-scale OOD detection.

| Method | AUROC |
|---|---|
| Supervised (OE) | 56.1 |
| RotNet | 65.3 |
| RotNet + Translation | 77.9 |
| RotNet + Self-Attention | 81.6 |
| RotNet + Translation + Self-Attention | 84.8 |
| RotNet + Translation + Self-Attention + Resize (Ours) | 85.7 |

Table 4: AUROC values of supervised and self-supervised OOD detectors. AUROC values are an average of 30 AUROCs corresponding to the 30 different models trained on exactly one of the 30 classes. Each model's in-distribution examples are from one of 30 classes, and the test out-of-distribution samples are from the remaining 29 classes. The self-supervised methods greatly outperform the supervised method. All values are percentages.

## 5 Conclusion

In this paper, we applied self-supervised learning to improve the robustness and uncertainty of deep learning models beyond what was previously possible with purely supervised approaches. We found large improvements in robustness to adversarial examples, label corruption, and common input corruptions. For all types of robustness that we studied, we observed consistent gains by supplementing current supervised methods with an auxiliary rotation loss. We also found that self-supervised methods can drastically improve out-of-distribution detection on difficult, near-distribution anomalies, and that in CIFAR and ImageNet experiments, self-supervised methods outperform fully supervised methods. Self-supervision had the largest improvement over supervised techniques in our ImageNet experiments, where the larger input size meant that we were able to apply a more complex self-supervised objective. Our results suggest that future work in building more robust models and better data representations could benefit greatly from self-supervised approaches.

## 5.1 Acknowledgments

This material is in part based upon work supported by the National Science Foundation Frontier Grant. Any opinions, findings, and conclusions or recommendations expressed in this material are those of the author(s) and do not necessarily reflect the views of the National Science Foundation.

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
