[Supplementary Material]

# Using Self-Supervised Learning Can Improve Model Robustness and Uncertainty Supplementary Materials

**Dan Hendrycks**
UC Berkeley
hendrycks@berkeley.edu

**Mantas Mazeika**[*]
UIUC
mantas3@illinois.edu

**Saurav Kadavath**[*]
UC Berkeley
sauravkadavath@berkeley.edu

**Dawn Song**
UC Berkeley
dawnsong@berkeley.edu

## A  Self-Supervised Learning for Multi-Class OOD Detection

| $\mathcal{D}_{in}$ | $\mathcal{D}_{out}^{test}$ | FPR95 ↓ | | AUROC ↑ | | AUPR ↑ | |
|---|---|---|---|---|---|---|---|
| | | MSP | Rotation | MSP | Rotation | MSP | Rotation |
| CIFAR-10 | Gaussian | 8.1 | 1.2 | 96.3 | 99.0 | 70.8 | 85.6 |
| | Rademacher | 5.9 | 1.1 | 97.5 | 99.1 | 79.4 | 86.3 |
| | Blobs | 13.3 | 2.3 | 94.6 | 98.9 | 68.3 | 86.5 |
| | Textures | 45.4 | 8.9 | 87.9 | 97.4 | 56.2 | 86.7 |
| | SVHN | 25.7 | 2.7 | 91.9 | 98.9 | 64.0 | 89.8 |
| | Places365 | 46.0 | 38.4 | 87.7 | 92.2 | 57.2 | 71.3 |
| | LSUN | 39.5 | 28.7 | 88.5 | 93.2 | 57.2 | 71.0 |
| | CIFAR-100 | 45.9 | 44.9 | 87.2 | 90.9 | 54.1 | 67.7 |
| | Mean | 28.7 | **16.0** | 91.4 | **96.2** | 63.4 | **80.6** |

Table 1: Out-of-distribution example detection results for the maximum softmax probability (MSP) baseline and our rotation method. All results are percentages and the average result of 5 runs.

The full multi-class out-of-distribution detection results are in Table 1. Auxiliary rotation prediction results in large improvements across the board for numerous anomaly types. In all cases, rotation prediction improves performance. This demonstrates that auxiliary rotation prediction is not only useful for one-class detection but can also augment detectors based on multi-class representations. For descriptions of metrics, we refer the reader to Hendrycks et al. [2019].

**OOD Datasets.**  For multi-class OOD detection, we evaluate our detectors on a wide variety of OOD data with CIFAR-10 as the in-distribution. *Gaussian* OOD data has each pixel sampled from an isotropic Gaussian distribution. *Rademacher* images have each pixel sampled IID from an Rademacher distribution, which takes values 1 and −1 with equal probability. *Blobs* images are algorithmically generated amorphous shapes with distinct edges. *Textures* is a dataset of describable texture images. *SVHN* is a dataset of house numbers extracted from Google Street View. *Places365* contains images for scene recognition instead of object recognition. *LSUN* is another scene understanding dataset that fewer classes than Places365 [Yu et al., 2015]. *CIFAR-100* is the 100-class counterpart to CIFAR-10. Importantly, the CIFAR-10 and CIFAR-100 classes do not overlap, so CIFAR-100 data is OOD with CIFAR-10 as the in-distribution.

---

[*]Equal Contribution.

## B    ImageNet OOD Dataset

The classes are 'acorn', 'airliner', 'ambulance', 'American alligator', 'banjo', 'barn', 'bikini', 'digital clock', 'dragonfly', 'dumbbell', 'forklift', 'goblet', 'grand piano', 'hotdog', 'hourglass', 'manhole cover', 'mosque', 'nail', 'parking meter', 'pillow', 'revolver', 'rotary dial telephone', 'schooner', 'snowmobile', 'soccer ball', 'stingray', 'strawberry', 'tank', 'toaster', and 'volcano'. These classes were selected so that there is no obvious overlap, unlike classes such as 'bee' and 'honeycomb.' There are 1,300 training images per class, and 100 test images per class. To create a dataset with 100 test images per class, we took ImageNet's 50 validation images, and we collected an additional 50 images for each class for an expanded test set. The data is available for download at <https://github.com/hendrycks/ss-ood>.

## C    Additional Ablations

**Not attacking the rotation branch.**    To gauge the effect of attacking the rotation branch during training, we retrain the auxiliary rotation method with the adversary only attacking the classification branch. We find this performs similarly to attacking both the classification and rotation branches, which indicates that the rotation loss itself is the crucial component.

**Comparison with rotation augmentation.**    Our results demonstrate myriad benefits of rotation prediction, so a natural baseline for comparison is rotation data augmentation. To this end, we retrain the baseline network from the common corruptions section and augment the dataset with rotations of multiples of 90 degrees. We find that this *decreases* average accuracy across corruptions from 72.3% to 63.7%. By contrast, training with auxiliary rotation prediction improves average accuracy to 76.9%.