[Reviews · NeurIPS 2019]

Reviewer 1



The authors present a way of self-supervised auxiliary learning in which the images in the training set are rotated with 4 different rotations, and the neural network has to predict the type of rotation. The authors show with various experiments that this type of SSL increases the robustness against all kinds of perturbations, ranging from adversarial attacks to motion blur and fog. In addition, the outputs indicating the rotation can be used for detecting outliers. The article makes a good case for both contributions. One main remark is that the title of the article talks about uncertainty estimation, while the experiments focus on outlier detection. These two tasks are related but not identical. Outlier detection boils down to binary classification, while uncertainty estimation produces continuous real values. In principle, the scheme introduced by the authors could be used to produce floating point numbers (the sum of outputs is such a number), but additional experiments should then show that the magnitude of this "uncertainty" then makes sense. One of the enticing properties of the method proposed by the authors is that it is so simple / elegant. When comparing with other approaches for OOD detection, the other approaches all have been explicitly made for this task. However, how well would it work to just take the max probability of the normal neural network (instead of the auxiliary outputs)? Should the probability of the max class also not be lower for outliers? And what about estimating the uncertainty via dropout at test time? How well does that work? Concerning robustness, I would wonder why the proposed auxiliary learning works. When also rotating the images, does this lead to more "blurry" features, so that it can deal with blur better? Differently put: Would it work for any auxiliary self-supervised task? What is special about rotations? Finally, when introducing this training, the "normal" performance almost always drops. For example in table 1, the performance goes from 94.8 to 83.5. What do the authors think of this matter? Can it be amended in the future, or is this just a trade-off between robustness and performance on clean samples? Overall, the article is well-written and makes a clear contribution to the field.

Reviewer 2



- This is an experimental paper with a quite clear message: The auxiliary rotation-based self-supervision may not improve accuracy, but it does something in robustness (both in adversarial, and corruption), and (especially) in OOD detection. - Despite of its simplicity, the idea is generally well-presented, with good experimental results. - However, I feel the current manuscript does not provide enough motivation or insight on why the auxiliary rotation task could improve robustness and OOD detection. More detailed analysis on the results would much help the readers to understand the significance of this work: e.g. ablation study, comparing characteristics of the original and proposed networks in the viewpoint of robustness/OOD. - My another concern is about a lack of novelty: Provided that "using pre-training can improve robust and uncertainty" [1], the results in this paper may not be that surprising for some readers, as self-supervision might be just another form of supervised bias in training a network. Again, I think this issue might be relaxed by providing more detailed analysis of the results: Is the claim generalizable to the other self-supervisions apart from the rotation? How about other auxiliary tasks, in the general framework of multi-task learning? Such questions should be justified. - L117 (Section 3.1): In the adv. robustness part, It seems the proposed method makes two (different) modifications from the original adv. training: adding SS-loss (a) on the training objective, and (b) on the PGD objective. Among those two, which one would be more critical for the observed gain? [1] Hendrycks et al., Using Pre-Training Can Improve Model Robustness and Uncertainty, ICML 2019. ----------After rebuttal---------- The authors may discover an interesting finding. But, even after reading the rebuttal, I am still unclear why predicting rotations improves robustness and uncertainty. They provide a part of reasoning/insight in the rebuttal letter, which is nice (hence, I increase my rate a bit, still on a negative side though), but not enough for me, e.g., then the authors should also show whether other non-rotational self-supervised learning works or not to support it. I think the paper has a good potential in the future, but not ready to publish as the current form.

Reviewer 3



Authors proposed using self-supervised learning methods to improve deep learning methods generalization properties such robustness against adversarial attacks and also to improve uncertainty estimates. Authors propose a simple extension to already established algorithm, i.e. PGD, by adding a term (second term in eq.3) to the loss. They show improvement of their proposed method in different problems e.g. out-of-distribution detection. They have used several different datasets e.g. Cifar, Imagenet. Paper is well written and experiments are conducted well. However, the novelty of the proposed method is limited and improvements are not surprising. Using self-supervised learning as a "data augmentation" method will improve the generalization performance of methods. Adding "extra" data points that are strictly not in original dataset will also improve the performance metrics. Hence in my opinion although paper is well written and experiments are well conducted the novelty of the paper is limited.

[Author Response · NeurIPS 2019]

Reviewers, thank you for your careful analyses of our paper.

We would like to clarify the value of our work for self-supervised learning, input corruption robustness, adversarial examples, label corruption robustness, and out-of-distribution detection. A preliminary version of this work has been well-received by the self-supervised community as one of four long oral presentations at a top self-supervised workshop. One reason for its positive reception is, in OOD detection, prior art on natural images for unsupervised techniques such as density estimators and one-class SVMs have performance near chance levels [1]. However, we show that five different self-supervised techniques straightforwardly improve over both unsupervised and one-class methods. We also show that a self-supervised multi-task combination can *even surpass fully supervised techniques* (see Table 4). Another reason our work is valuable to the self-supervised learning community is because we identify self-attention as a useful architectural change; this finding is valuable because self-supervised advancements greatly depend on researchers identifying appropriate architectural choices [2]. For these reasons and more, we believe our OOD detection results have clear value to both the self-supervised learning and OOD detection research communities.

Regarding robustness, training against more data generally does not improve corruption robustness—even training against different corrupted data does not improve robustness [3,4]. These previous works show that training against one type of corruption does not confer robustness to novel corruptions. However, we find self-supervised learning does improve robustness to various novel corruptions. Moreover, we independently experimented with pre-training (R2) on ImageNet and found it did not improve corruption robustness. Further, while self-supervision may be thought of as "a 'data augmentation' method," augmenting the dataset with rotations of multiples of 90 degrees actually *decreases* corruption robustness from 72.3% to 63.7%, but with a rotation prediction loss, it improves to 76.9%. It was not obvious from prior work that combining fully supervised and self-supervised objectives could improve corruption robustness. Hence, this result is surprising and of value.

We agree with R2 that self-supervised learning is a form of bias or regularization, but whether this inductive bias helps is unclear a priori. For adversarial examples, there is much work on training with orders of magnitude more data to increase adversarial robustness [5,6,7]. Rather than training on significantly more data, we show it is possible to extract more predictive information from the training data with self-supervised learning. More, in many domains one does not have external data to train on, such as the medical domain. In these domains, improvements to label corruption robustness and adversarial robustness from self-supervised learning are especially valuable.

Figure 1: Predicting rotations requires shape, as texture alone is not sufficient for prediction.

R1 and R2 ask why predicting rotations improves robustness. Due to space constraints, we did not speculate on this in the paper, but we think part of the reason is that it requires modeling shape. For example, predicting the zebra's rotation in Figure 1 requires modeling contours and not just texture. This can lead to more robust representations. We will include discussion of this and further analysis in the updated draft.

**Individual responses**

R1: For OOD detection, our work focuses on the challenging one-class setting, meaning that we fix a single class as in-distribution and the rest as out-of-distribution. Thus, the MSP detector, MC-Dropout, and other techniques suited for multiclass do not apply since we learn with in-distribution data. The performance drop on clean data in Table 1 is a pervasive and a recognized shortcoming of adversarial training itself [8]. Finally, when rotating images, we use the rot90 function from NumPy. This avoids blurriness caused by resampling.

R2: We address many of the concerns in the general comments. On L117, you suggested not attacking the rotation branch, which is a good suggestion. We find that it interestingly performs similarly to attacking the rotation branch and will include this ablation in the updated draft. Thank you.

All our experiments were run with fixed random seeds and hyperparameters chosen as standard values or tuned on validation data. The computational cost of our experiments is high, but we agree that error bars are feasible and informative to add for the common corruption experiments. Due to your suggestion, we have now run these numerous experiments and will add error bars to the updated draft of the paper.

R3: We address many of the concerns in the general comments. In addition to our novel method from the OOD section, our main novelty is in our successful integration of self-supervised learning to four highly researched areas, and our demonstration that robustness and uncertainty can be new dimensions with which to judge self-supervised learning advancements. We will make this clearer in the paper. Thank you.

[1] Implicit Generation and Generalization with Energy Based Models. [2] Revisiting Self-Supervised Visual Representation Learning. [3] Comparing deep neural networks against humans: object recognition when the signal gets weaker. [4] Examining the Impact of Blur on Recognition by Convolutional Networks. [5] Are Labels Required for Improving Adversarial Robustness? [6] Adversarially Robust Generalization Just Requires More Unlabeled Data. [7] Unlabeled Data Improves Adversarial Robustness. [8] Adversarial Training Can Hurt Generalization.


[Meta-Review · NeurIPS 2019]

This paper received mixed reviews. All reviewers found the empirical findings in the paper to be very interesting. The main concern from reviewers was about the lack of theoretical justification for the findings. However, many empirical results precede theoretical results, and this paper's empirical results are interesting in its own right. The area chair has read the paper in detail. The paper is well written, and provides important empirical analysis for two timely questions in the field today: model robustness and self-supervised learning. The paper's many experiments on accuracy, out-of-distribution detection, and robustness make this work potentially very interesting to the research community. Although the techniques in the paper are straightforward, the empirical results establish a novel (to AC and reviewers knowledge) link between self-supervised learning and model robustness. The AC recommends acceptance.